# The CLAVATA receptor FASCIATED EAR2 responds to distinct CLE peptides by signaling through two downstream effectors

**Byoung Il Je[1,2†], Fang Xu[1†], Qingyu Wu[1], Lei Liu[1], Robert Meeley[3], Joseph P Gallagher[4], Leo Corcilius[5], Richard J Payne[5], Madelaine E Bartlett[4], David Jackson[1]***

[1]Cold Spring Harbor Laboratory, Cold Spring Harbor, United States; [2]Department of Horticultural Bioscience, College of Natural Resource and Life Science, Pusan National University, Miryang, Republic of Korea; [3]DuPont Pioneer, Agricultural Biotechnology, Johnston, United States; [4]University of Massachusetts Amherst, Amherst, United States; [5]The University of Sydney, Camperdown, Australia

**Abstract** Meristems contain groups of indeterminate stem cells, which are maintained by a feedback loop between *CLAVATA* (*CLV*) and *WUSCHEL* (*WUS*) signaling. CLV signaling involves the secretion of the CLV3 peptide and its perception by a number of Leucine-Rich-Repeat (LRR) receptors, including the receptor-like kinase CLV1 and the receptor-like protein CLV2 coupled with the CORYNE (CRN) pseudokinase. CLV2, and its maize ortholog FASCIATED EAR2 (FEA2) appear to function in signaling by CLV3 and several related CLV3/EMBRYO-SURROUNDING REGION (CLE) peptide ligands. Nevertheless, how signaling specificity is achieved remains unknown. Here we show that FEA2 transmits signaling from two distinct CLE peptides, the maize CLV3 ortholog ZmCLE7 and ZmFON2-LIKE CLE PROTEIN1 (ZmFCP1) through two different candidate downstream effectors, the alpha subunit of the maize heterotrimeric G protein COMPACT PLANT2 (CT2), and ZmCRN. Our data provide a novel framework to understand how diverse signaling peptides can activate different downstream pathways through common receptor proteins.
DOI: https://doi.org/10.7554/eLife.35673.001

**\*For correspondence:**
jacksond@cshl.edu

[†]These authors contributed equally to this work

**Competing interests:** The authors declare that no competing interests exist.

## Introduction

Stem cell proliferation and differentiation throughout plant life is regulated by a feedback loop between the homeodomain transcription factor *WUS* and *CLV* ligand-receptor signaling (*Mayer et al., 1998*; *Brand et al., 2000*; *Schoof et al., 2000*; *Yadav et al., 2011*; *Daum et al., 2014*). The secretion of the diffusible glycopeptide CLV3 from the central zone (CZ) stem cells of the SAM is believed to initiate signaling through LRR receptors (*Fletcher et al., 1999*; *Rojo et al., 2002*; *Kondo et al., 2006*; *Ohyama et al., 2009*; *Nimchuk et al., 2011b*), which transmits the signal to restrict the expression of WUS in the organizing center (OC) cells. To balance this system, WUS non-cell-autonomously promotes stem cell fate by activation of *CLV3* expression (*Yadav et al., 2011*; *Daum et al., 2014*). CLV3 is thought to be perceived by multiple receptor kinase and receptor like proteins, including the CLV1 LRR receptor-like kinase (*Clark et al., 1993*; *Clark et al., 1997*; *Brand et al., 2000*; *Ogawa et al., 2008*) and the related BAM receptors (*DeYoung et al., 2006*; *Deyoung and Clark, 2008*; *Nimchuk et al., 2015*; *Shinohara and Matsubayashi, 2015*), or by a heterodimer of the receptor like protein CLV2 and the transmembrane pseudokinase CRN (*Kayes and Clark, 1998*; *Jeong et al., 1999*; *Miwa et al., 2008*; *Müller et al., 2008*; *Bleckmann et al., 2010*;

**eLife digest** Like animals, plants are made up of many different types of cells, which descend from undifferentiated cells called stem cells. Thanks to these cells, plants are able to grow and develop throughout their lives. Stem cells live at the tips of the plant's shoots and roots. They constantly divide to produce new cells to self-renew or replace specific plant cells in need of repair. Over time, they change – or differentiate – to go on to become part of tissues like leaves, roots, stems, shoots, flowers or fruits.

To maintain a continuous pool of undifferentiated stem cells and to make sure that stem cells divide at the correct pace, neighbouring cells emit signals that control the activity of stem cells. The new stem cells that remain close to these 'maintenance signals' continue to behave like stem cells, but those displaced away begin to differentiate. Stem cells can receive many different types of signals, but how are these signals filtered and passed onto different places within the cell?

To test this, Je, Xu et al. created maize plants that contained mutations in a number of known signalling molecules to see if these molecules used the same communication pathway. The results showed that stem cells could integrate the different signals. Even if the signals pass through the same receiver (a receptor protein called FASCIATED EAR2), each signal exits the receptor as a different message, and attaches to a different messenger protein to relay specific information about stem cell maintenance to the cell.

A next step will be to test if other plants use the same signalling pathways in the same ways to send messages between cells. A better knowledge about stem cell signals in plants could help to develop more productive crops. Previous work has found that precise control of stem cell pathways can help breed crops with more seeds or bigger fruits. These kinds of changes have been selected naturally by humans since the dawn of civilization, but we need to accelerate these advances to help meet the needs of the growing world population and improve agricultural sustainability.

DOI: https://doi.org/10.7554/eLife.35673.002

*Zhu et al., 2010*; *Nimchuk et al., 2011a*), or by the receptor-like kinase RPK2 (*Mizuno et al., 2007*; *Nodine et al., 2007*; *Kinoshita et al., 2010*). The relationship between CLV1 and CLV2 is not clear-CLV1 can form homodimers, or higher order complexes with CLV2/CRN, to signal co-operatively in the SAM (*Guo et al., 2010*; *Somssich et al., 2015*), but it seems that CLV2/CRN is not essential for CLV3 perception or for CLV1 signaling (*Müller et al., 2008*; *Nimchuk et al., 2011b*; *Nimchuk, 2017*). In contrast to CLV1, CLV2 does not bind CLV3 peptide directly (*Shinohara and Matsubayashi, 2015*), and its expression is not restricted to the SAM, suggesting that it might function as a co-receptor in additional pathways beyond CLV3 signaling. Indeed, CLV2 appears to be involved in signaling by several CLE peptides (*Fiers et al., 2005*; *Meng and Feldman, 2010*; *Hazak et al., 2017*) and in biotic interactions (*Replogle et al., 2011*; *Hanemian et al., 2016*), suggesting it plays diverse functions in plant development and immunity (*Pan et al., 2016*). The multiple roles of CLV2 promote the question of how it confers signal specificity. Two candidate downstream effectors of CLV2 have been identified. One is the transmembrane pseudokinase CRN, discovered in *Arabidopsis,* and the second is COMPACT PLANT2 (CT2), the heterotrimeric G protein alpha subunit, discovered in maize (*Bommert et al., 2013a*). However, since CRN and CT2 were identified in different species, their molecular and genetic interactions remain unknown.

The CLV-WUS pathway is widely conserved (*Somssich et al., 2016*; *Soyars et al., 2016*). In maize, *THICK TASSEL DWARF1* (*TD1*) and *FEA2* are *CLV1* and *CLV2* orthologs, and function similarly to restrict inflorescence shoot meristem proliferation (*Taguchi-Shiobara et al., 2001*; *Bommert et al., 2005*). Two maize WUS orthologs, ZmWUS1 and ZmWUS2, have been predicted by phylogenetic analysis, and a *ZmWUS1* reporter is expressed in the presumptive organizing center of the inflorescence shoot meristem (*Je et al., 2016*), but these genes have not been functionally characterized (*Nardmann and Werr, 2006*). In rice, *FLORAL ORGAN NUMBER 1* (*FON1*), the *CLV1* ortholog, and *FON2,* the *CLV3* ortholog, similarly function in floral development in a common pathway, as expected (*Suzaki et al., 2004*; *Chu et al., 2006*; *Suzaki et al., 2006*; *Suzaki et al., 2008*; *Suzaki et al., 2009*), whereas a second rice CLE peptide gene, *FON2-LIKE CLE PROTEIN1* (*FCP1*) controls stem cell proliferation independent of *FON1* (*Suzaki et al., 2008*). The rice WUS homolog,

*TILLERS ABSENT1/MONOCULM3* functions in axillary shoot meristem formation (*Tanaka et al., 2015*; *Lu et al., 2015*), and WUS function in the shoot apical meristems appears to have been taken over by the *WUSCHEL RELATED HOMEOBOX4* (*WOX4*) gene (*Ohmori et al., 2013*).

How specificity is achieved is a common question in signal transduction pathways. Recently, we identified a distinct CLV receptor, *FASCIATED EAR3 (FEA3)* in maize and *Arabidopsis*, and found that FEA3 controls responses to the maize FCP1 (ZmFCP1) CLE peptide (*Je et al., 2016*). Here, we show that the maize CLV2 ortholog FEA2 also participates in ZmFCP1 signaling, in addition to controlling responses to the maize CLV3 ortholog, ZmCLE7 (*Je et al., 2016*). To ask how specificity from these different CLE peptide inputs is achieved, we first isolated mutant alleles of the maize *CRN* gene. Consistent with results in *Arabidopsis* (*Miwa et al., 2008*; *Müller et al., 2008*; *Bleckmann et al., 2010*; *Zhu et al., 2010*; *Nimchuk et al., 2011a*), we found that *fea2* was epistatic to *Zmcrn* in control of meristem size, but *Zmcrn;ct2* double mutants showed an additive enhanced phenotype, suggesting they act in parallel pathways, despite the fact that FEA2 binds both ZmCRN and CT2 in co-immunoprecipitation (co-IP) experiments. Strikingly, *ct2* and *Zmcrn* mutants were resistant to different CLE peptides, ZmCLE7 and ZmFCP1, respectively, but *fea2* was resistant to both, suggesting that FEA2 controls responses to different CLE peptides by acting through different downstream effectors.

## Results

### Both *fea3* and *fea2* mutants are resistant to the ZmFCP1 peptide

We recently described a new CLE signaling pathway in maize, in which ZmFCP1 peptide signals through FEA3 to restrict ZmWUS1 expression from below its organizing center expression domain (*Je et al., 2016*). To test this model, we used a 2-component transactivation system (*Wu et al., 2013*; *Je et al., 2016*) to drive ZmFCP1 expression in developing primordia, below the ZmWUS1 domain (*Je et al., 2016*; *Nardmann and Werr, 2006*). As previously described, this expression reduced meristem size of wild type SAMs (*Je et al., 2016*), however we found that meristem size was only partially rescued when *ZmFCP1* expression was transactivated in a *fea3* mutant background (*Figure 1A and B*), suggesting that ZmFCP1 signals through additional receptors. We therefore conducted peptide response assays using *fea2* mutants, and found that they were also insensitive to ZmFCP1 peptide treatment, as well as to ZmCLE7, the maize CLV3 ortholog (*Figure 1C*) (*Je et al., 2016*). Interestingly, *fea2;fea3* double mutants restored the size of ZmFCP1 treated meristems to control levels, suggesting that ZmFCP1 signaling is transmitted predominantly through both FEA2 and FEA3 (*Figure 1D*). *fea3* mutants are resistant only to ZmFCP1, and not to ZmCLE7 (*Je et al., 2016*), so we next asked how FEA2 might transmit signals from different CLE peptides.

### *Zmcrn* mutants are fasciated

In maize, FEA2 signals through CT2, the alpha subunit of the heterotrimeric G protein (*Bommert et al., 2013a*), but in *Arabidopsis* the FEA2 ortholog CLV2 is thought to signal through a membrane bound pseudokinase, CRN (*Miwa et al., 2008*; *Müller et al., 2008*; *Bleckmann et al., 2010*; *Zhu et al., 2010*; *Nimchuk et al., 2011a*). To ask if CRN also functions in CLV signaling in maize, we identified maize CRN (ZmCRN) by phylogenic analysis (*Figure 2—figure supplement 1A*). As is the case for *Arabidopsis CRN*, *ZmCRN* was also predicted to encode an inactive pseudokinase (*Figure 2—figure supplement 1B*) (*Boudeau et al., 2006*; *Nimchuk et al., 2011a*). We identified a predicted null allele as a *Mu* transposon insertion from the Trait Utility System in Corn (TUSC) resource (*McCarty and Meeley, 2009*), 52 bp downstream of the predicted translation start site (*Figure 2A*). We backcrossed this *Mu* insertion line three times to the standard B73 inbred line, and dissected homozygous mutant or normal sib samples for meristem analysis. The maize *crn* (*Zmcrn*) mutants had larger vegetative shoot meristems (130.0 ± 4.1 µm, compared to 109.2 ± 4.6 µm for normal sibs, P value < 0.0001, two-tailed t test, *Figure 2B and C*), and developed fasciated ear primordia with enlarged and split inflorescence meristems (*Figure 2D*), reminiscent of other *fasciated ear* mutants (*Taguchi-Shiobara et al., 2001*; *Bommert et al., 2005*; *Bommert et al., 2013a*; *Je et al., 2016*). Concurrently, we identified a second candidate allele by map-based cloning of a fasciated mutant, *fea*148* (*Figure 2—figure supplement 2A*), from an ethyl methyl sulfonate (EMS) screen in the B73 background (hereafter *Zmcrn-148*). *Zmcrn-148* introduced a stop codon within the

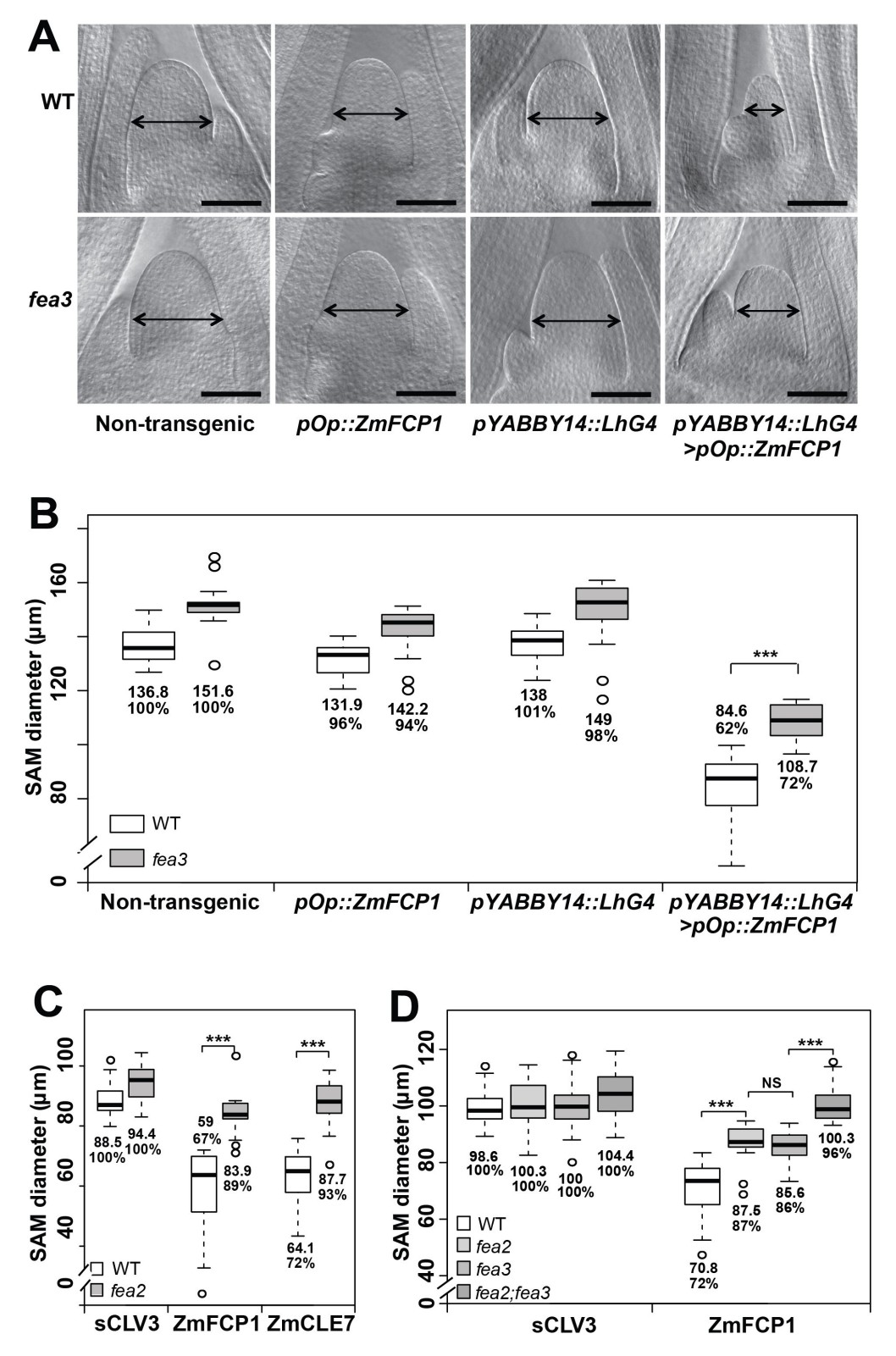

**Figure 1.** Both *fea3* and *fea2* mutants are resistant to ZmFCP1 peptide. (**A**) Transactivation of ZmFCP1 in primordia using a pYABBY14:LhG4 driver led to a strong reduction in vegetative SAM size as compared to a non-transgenic control, but this effect was only partially rescued in a *fea3* mutant background; SAM diameter was quantified (**B**). In CLE peptide treatments, *fea2* mutants were resistant to ZmFCP1, as well as to ZmCLE7 (**C**), and *fea3; fea2* double mutants showed additive resistance to ZmFCP1, restoring SAM size to normal (**D**). Scale bars; 100 µm in A. n = 20 (**B, C**) and 30 (**D**) plants

*Figure 1 continued on next page*

*Figure 1 continued*

for each genotype. Data in B, C and D are shown by box plots. The mean values as well as the relative % to each untreated control are listed for each genotype. The untreated controls are set to 100%: '***': P value < 0.0001, two-tailed, two-sample t test. 'NS': not significant.

DOI: https://doi.org/10.7554/eLife.35673.003

The following source data is available for figure 1:

**Source data 1.** CLE peptide treatments of *fea2;fea3* a segregating population.

DOI: https://doi.org/10.7554/eLife.35673.004

predicted pseudokinase domain (*Figure 2A*), and plants homozygous for this mutation developed a similar fasciated ear phenotype (*Figure 2—figure supplement 2C*). We next crossed heterozygous *Zmcrn-148* plants with *Zmcrn* mutants. The F1 plants developed fasciated ears, while *Zmcrn/+* or *Zmcrn-148 /+* heterozygotes had normal ear primordia, suggesting that these mutations were allelic (*Figure 2—figure supplement 3*), and confirming that *CRN* functions in shoot meristem size control in maize, similar to its role in *Arabidopsis. ZmCRN* was expressed throughout the SAM and more strongly in the peripheral domain and leaf primordia (*Figure 2E*, confirmed by laser capture micro-dissection RNAseq, *Figure 2—figure supplement 4*). Next, since *fea2* and other *fea* mutants are associated with quantitative variation in kernel row number (KRN) (*Bommert et al., 2013b*), we took advantage of the identification of *ZmCRN* to ask if it was also associated with this yield trait. We conducted a candidate gene association study using a maize association panel of 368 diverse inbred lines (*Li et al., 2013*; *Liu et al., 2015*). We found that three SNPs in the 3'UTR region of *CRN* showed significant association with KRN in multiple environments, below the threshold p-value<0.001 (*Figure 2—figure supplement 5* and *Supplementary File 1*). These results suggest that natural variation in *ZmCRN* may underlie subtle variation in inflorescence meristem size sufficient to enhance KRN, with the potential to benefit maize yields.

## *ZmCRN* and *FEA2* function in a common pathway

In *Arabidopsis*, CRN is thought to signal downstream of CLV2 and correspondingly the double mutants show an epistatic interaction (*Müller et al., 2008*). To ask if this relationship was conserved in maize, we measured the SAM size in a segregating double mutant population. As expected, both *Zmcrn* and *fea2* vegetative meristems were larger than normal (166.3 ± 8.3 μm, or 176.1 ± 9.8 μm respectively, compared to 139.7 ± 4.8 μm for normal sibs, P value < 0.0001, two-tailed t test, *Figure 3A and B*), and the *Zmcrn; fea2* double mutants (177.2 ± 13.3 μm) were similar to the *fea2* single mutants (176.1 ± 9.8 μm, P value = 0.68, two-tailed t test) (*Figure 3A and B*). We also characterized ear inflorescence meristems and found that *fea2* had stronger fasciated ears than those of *Zmcrn*, but the double mutants resembled *fea2* single mutants (*Figure 3C*). Together, these results indicate that *fea2* is epistatic to *Zmcrn*, suggesting that *FEA2* and *ZmCRN* function in a common pathway in maize, as in *Arabidopsis*.

## *ZmCRN* and *CT2* function in different pathways

We next asked if *ZmCRN* and *CT2* function in the same or in different pathways, again by double mutant analysis. Both *Zmcrn* and *ct2* mutants had larger SAMs compared with their normal sibs (161.5 ± 10.6 μm, or 157.1 ± 11.8 μm respectively, compared to 139.7 ± 8.5 μm for normal sibs, P value < 0.0001, two-tailed t test, *Figure 3D and E*), but the SAMs of double mutants were significantly larger than each single mutant (191.8 ± 18.6 μm, P value < 0.0001, two-tailed t test, *Figure 3D and E*), suggesting an additive interaction. *Zmcrn; ct2* double mutant ear inflorescences also showed additive enhancement in fasciation, compared to each single mutant (*Figure 3F*), confirming the additive interaction between *ct2* and *Zmcrn*. In summary, double mutant analyses and quantification of meristem sizes indicated that *ZmCRN* functions in the same pathway as *FEA2* and, as previously reported, *CT2* also functions in the same pathway as *fea2* (*Bommert et al., 2013a*), but *CT2* and *ZmCRN* themselves function in different pathways. This result is most easily explained by the hypothesis that *FEA2* functions in two different pathways, one with *CT2* and a second with *ZmCRN*.

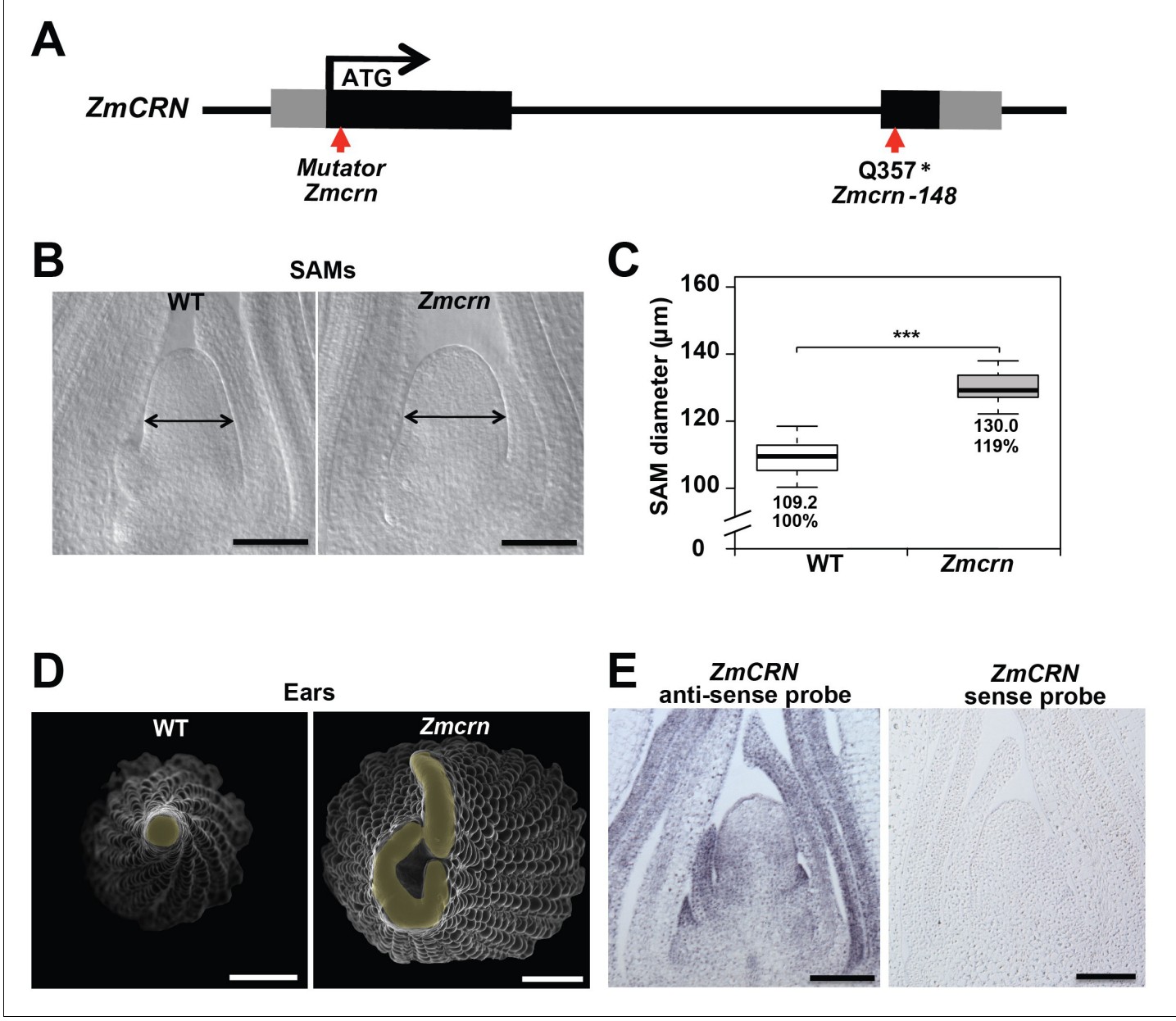

**Figure 2.** *Zmcrn* mutants develop fasciated ears. (A) Schematic of the *Zmcrn* mutant alleles. The arrows indicate the position of the Mutator transposon insertion and premature stop codon mutation. (B) Cleared SAMs from wild type (WT) and *Zmcrn* plants. The *Zmcrn* SAM has a larger diameter (double-headed arrows), SAM diameter was quantified (C). (D) Scanning electron microscopy images of WT and *Zmcrn* ear primordia (inflorescence meristems in yellow). The *Zmcrn* ear shows an enlarged and fasciated inflorescence meristem. (E) In-situ hybridization showing *ZmCRN* expression throughout the SAM, with higher expression in peripheral zone and leaf primordia. Scale bar: 100 μm in B and E, 500 μm in D. n = 30 (C) plants for each genotype. Data in C are shown by box plots. The mean values as well as the relative % to the WT control are listed. '***': P value < 0.0001, two-tailed, two-sample t test.

DOI: https://doi.org/10.7554/eLife.35673.005

The following source data and figure supplements are available for figure 2:

**Source data 1.** SAM size of *Zmcrn* in a segregating population.
DOI: https://doi.org/10.7554/eLife.35673.011
**Figure supplement 1.** Phylogeny of CRN related proteins, and ZmCRN features of a pseudokinase.
DOI: https://doi.org/10.7554/eLife.35673.006
**Figure supplement 2.** Mapping of the *fea*148* fasciated ear mutant.
DOI: https://doi.org/10.7554/eLife.35673.007

*Figure 2 continued*

**Figure supplement 3.** *Zmcrn/Zmcrn-148* F1 plants develop fasciated ears.
DOI: https://doi.org/10.7554/eLife.35673.008
**Figure supplement 4.** Expression of *FEA2*, *CT2* and *ZmCRN* in different domains of the SAM.
DOI: https://doi.org/10.7554/eLife.35673.009
**Figure supplement 5.** The association of *ZmCRN* locus with kernel row number (KRN).
DOI: https://doi.org/10.7554/eLife.35673.010

## FEA2 interacts physically with CT2 and with ZmCRN

To test the two-pathway hypothesis, we tested protein-protein interactions using Co-IP assays. We used an internal YFP fusion of CT2 that we previously found to be biologically active (*Bommert et al., 2013a*), and C terminal mCherry or Myc fusions of ZmCRN or FEA2, respectively, which are predicted to be correctly localized and active, based on similar fusions (*Bleckmann et al., 2010*; *Nimchuk, 2017*). We first confirmed the expected plasma membrane localization of ZmCRN-mCherry by transient expression and plasmolysis (*Figure 4A*), consistent with FEA2 and CT2 localization (*Bommert et al., 2013a*). ZmCRN-mCherry also co-localized with FEA2-YFP and CT2-YFP on the plasma membrane when they were co-expressed (*Figure 4—figure supplement 1*). We then tested pairwise interactions using Co-IP experiments following transient expression. ZmCRN-mCherry was able to pull down FEA2-Myc, but not CT2-YFP, even when FEA2-YFP was also co-expressed (*Figure 4B*). We confirmed that CT2-YFP was properly expressed, because it could pull down FEA2-Myc (*Figure 4C*), as previously demonstrated by *in vivo* co-IPs (*Bommert et al., 2013a*). To validate these interactions, a reciprocal Co-IP experiment was carried out, in which all three proteins were co-expressed, and we consistently found that FEA2-Myc could IP CT2-YFP or ZmCRN-mCherry (*Figure 4D*), further confirming that FEA2 formed complexes with both CT2 and ZmCRN. As an independent test, we also used an optimized BiFC system, with monomeric Venus (mVenus) split at residue 210 to reduce background due to false positive interactions (*Gookin and Assmann, 2014*). We detected YFP signal when FEA2 fused with the N terminal part of mVenus (NmVen210) was co-expressed with ZmCRN fused with the C terminal part (CmVen210) (*Figure 4—figure supplement 2*), confirming a direct interaction between FEA2 and ZmCRN. Similar results were reported in Arabidopsis using BiFC to detect CRN-CLV2 interactions (*Zhu et al., 2010*). However, we failed to detect a YFP signal when FEA2-NmVen210 was co-expressed with CT2-CmVen210 (*Figure 4—figure supplement 2*). The interaction between FEA2 and CT2 is well documented in maize by in vivo Co-IP experiments (*Bommert et al., 2013a*), and a failure to detect the same interaction using BiFC suggests that their interaction might be indirect, such as in a complex where their interaction is bridged by other protein(s). Lastly, as expected, no signal was detected when CT2-NmVen210 was co-expressed with ZmCRN-CmVen210 (*Figure 4—figure supplement 2*), confirming out Co-IP results, and supporting the hypothesis that they do not interact. The FEA2-ZmCRN and FEA2-CT2 interactions appeared to be quite stable, and were not affected by co-infiltration of CLE peptides (*Figure 4—figure supplement 3*).

In summary, the FEA2 receptor-like protein interacted with both candidate signaling molecules, ZmCRN and CT2, but these interactions appeared to be in different protein complexes, rather than in a common complex, because ZmCRN was not able to immunoprecipitate CT2.

## *ct2* and *Zmcrn* show differential sensitivity to ZmCLE7 and ZmFCP1 peptides

The activity of CLE peptides can be assayed using synthetic peptide treatments, which suppress the growth of the SAM and root apical meristem (*Ito et al., 2006*; *Kondo et al., 2006*). We therefore tested the sensitivity of each mutant to different CLE peptides, using embryo culture, as previously described (*Bommert et al., 2013a*; *Je et al., 2016*). *ct2* or *Zmcrn* segregating populations were grown in the presence of different peptides, and shoots fixed and cleared for SAM measurements after 12 days. We found that *ct2* mutants were partially resistant to ZmCLE7, but not to ZmFCP1 peptide (*Figure 5A and B*), suggesting that CT2 functions specifically in signaling by ZmCLE7, the maize CLV3 ortholog. In contrast, we found that *Zmcrn* mutants were partially resistant to ZmFCP1, but not to ZmCLE7 (*Figure 5C and D*), suggesting that ZmCRN functions specifically in a ZmFCP1

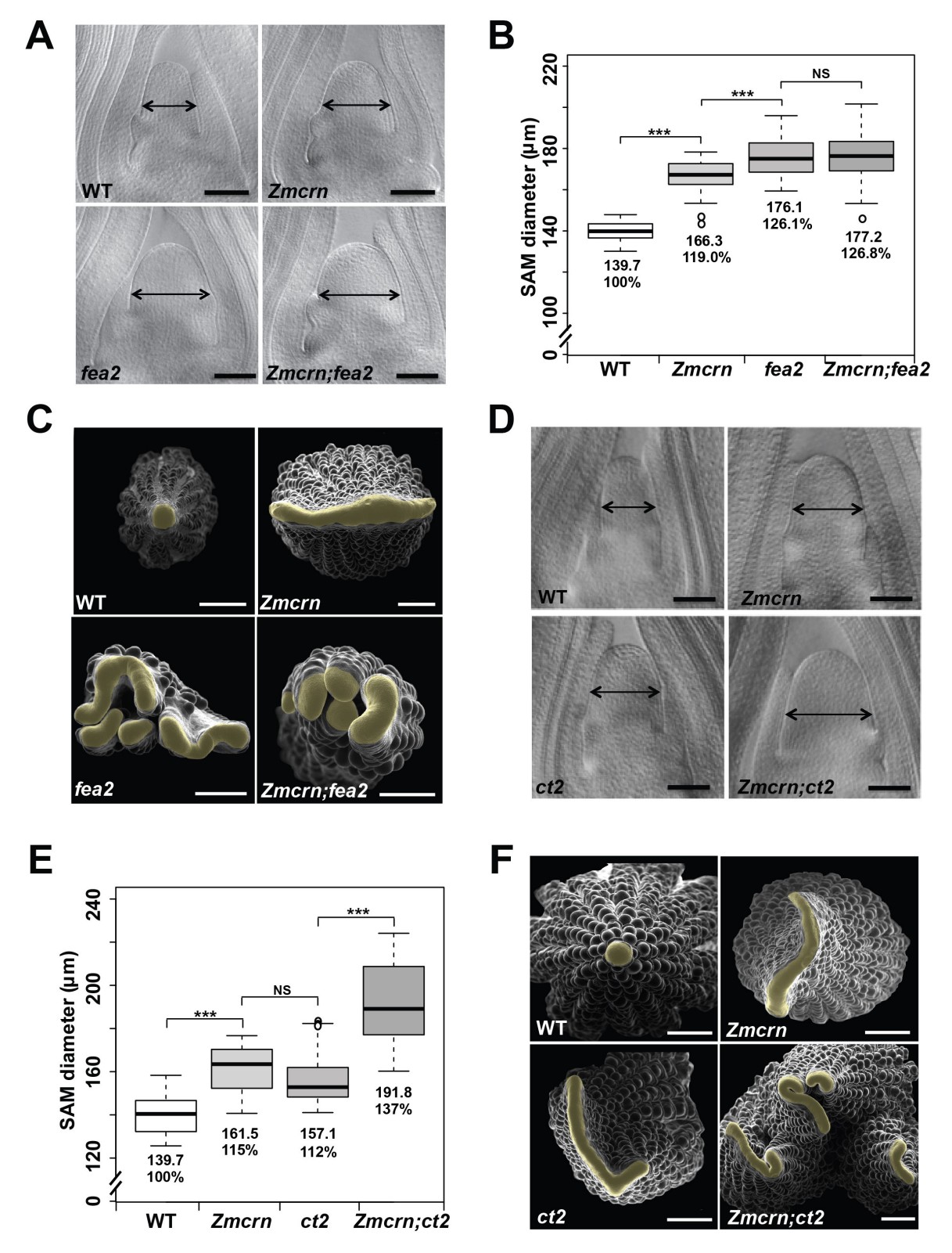

**Figure 3.** ZmCRN acts in a common pathway with FEA2, but not with CT2. (A) Cleared SAMs from wild type (WT), *Zmcrn*, *fea2*, and *Zmcrn;fea2* double-mutant plants. SAMs from *Zmcrn* and *fea2* plants were significantly wider than in wild type (double-headed arrows), but SAM size was not significantly different between *fea2* and *Zmcrn;fea2* double mutants, SAM diameter was quantified (B). (C) Ear meristems of *fea2;Zmcrn* double mutants resemble *fea2* single mutants. (D) Cleared SAMs from wild type, *Zmcrn*, *ct2*, and *Zmcrn;ct2* double-mutant plants. SAMs from *Zmcrn* and *ct2* plants were

*Figure 3 continued on next page*

*Figure 3 continued*

significantly wider than in wild type, and were additively increased in *Zmcrn;ct2* double mutants; SAM diameter was quantified (**E**). (**F**) *Zmcrn;ct2* double mutants had enhanced fasciation of ear primordia. Scale bars: 100 µm in A and D, 500 µm in C and F. n = 30 (**B, E**) plants for each genotype. Data in B and E are shown by box plots. The mean values as well as the relative % to the WT control are listed. '***': P value < 0.0001, two-tailed, two-sample t test, 'NS': not significant.

DOI: https://doi.org/10.7554/eLife.35673.012

The following source data is available for figure 3:

**Source data 1.** SAM size of *Zmcrn;fea2* in a segregating population.

DOI: https://doi.org/10.7554/eLife.35673.013

signaling pathway. To confirm these results, we treated each mutant with both ZmCLE7 and ZmFCP1 together. We found that only *fea2,* but not *ct2* or *Zmcrn* mutants, showed resistance to the double peptide treatment (*Figure 5E and F*). Together, these results suggest that FEA2 functions in both ZmCLE7 and ZmFCP1 signaling pathways, but CT2 and ZmCRN function specifically in ZmCLE7 or in ZmFCP1 signaling, respectively. As FEA3 also acts to transmit the ZmFCP1 signal (*Je et al., 2016*), we used genetic analysis to ask if ZmCRN also functions downstream of FEA3. In a segregating double mutant population, the SAMs of *Zmcrn* and *fea3* mutants were both larger than normal, as expected (160.2 ± 6.7 µm, or 176.8 ± 8.2 µm respectively, compared to 142.6 ± 6.0 µm for normal sibs, P value < 0.0001, two-tailed t test), and the *fea3; Zmcrn* double mutants were larger than the single mutants (221.5 ± 21.2 µm, P value < 0.0001, two-tailed t test), suggesting that FEA3 and ZmCRN do not function in a common pathway (*Figure 5—figure supplement 1A and B*). Similar findings were observed for *fea3; ct2* double mutants (*Figure 5—figure supplement 1C and D*), suggesting that FEA3 and CT2 also do not function in a common pathway. Thus, ZmFCP1 signaling appears to be mediated by two different pathways, one acting through FEA2 coupled with ZmCRN, and another acting through FEA3 working through as yet unknown downstream component (s).

In summary, through identification of maize *crn* mutants, we were able to show that signaling through FEA2 by two different CLE peptides is differentiated using different candidate downstream signaling components; with the ZmCLE7 signal passing through CT2 and the ZmFCP1 signal passing through ZmCRN (*Figure 6*).

## Discussion

A major question in signal transduction is how multiple inputs can be translated into distinct outputs. CLV-WUS feedback signaling is the central regulatory pathway in shoot meristem development, and perception of CLV3 peptide involves the CLV1 receptor-like kinase and the CLV2 receptor-like protein together with the CRN pseudokinase (*Brand et al., 2000*; *Schoof et al., 2000*; *Miwa et al., 2008*; *Müller et al., 2008*; *Bleckmann et al., 2010*; *Zhu et al., 2010*; *Nimchuk et al., 2011a*). However, genetic evidence in both maize and *Arabidopsis* suggests these receptors function independently, and CLV2, and its maize ortholog FEA2, respond to multiple CLE peptides (*Bommert et al., 2005*; *Fiers et al., 2005*; *Müller et al., 2008*; *Guo et al., 2010*; *Meng and Feldman, 2010*; *Je et al., 2016*; *Hazak et al., 2017*). So how is the information conferred by these different signals kept separate during transmission through a common receptor?

To address this question and further decipher the FEA2 signaling pathway, we isolated mutants in the maize *CRN* ortholog, *ZmCRN,* by reverse genetics and by cloning a newly identified fasciated ear mutant *fea*148. ZmCRN was predicted to encode a membrane localize pseudokinase, like *CRN* in *Arabidopsis* (*Nimchuk et al., 2011a*), and characterization of the mutants indicated that *ZmCRN* similarly functioned as a negative regulator of stem cell proliferation. We found that *fea2* was epistatic to *Zmcrn* and that FEA2 and ZmCRN interacted directly, using Co-IP and BiFC assays of proteins transiently overexpressed in *N. benthamiana,* suggesting that ZmCRN is a signaling component in the FEA2 pathway. Natural variation in the CLV-WUS pathway underlies yield improvements in different crop species including tomato, maize and mustard (*Bommert et al., 2013b*; *Fan et al., 2014*; *Xu et al., 2015*; *Je et al., 2016*), and FEA2 is a quantitative trait locus (QTL) for kernel row number (KRN) (*Bommert et al., 2013b*). In this study, we used a maize association panel of 368 diverse inbred lines to show that *ZmCRN* also had significant association with KRN under

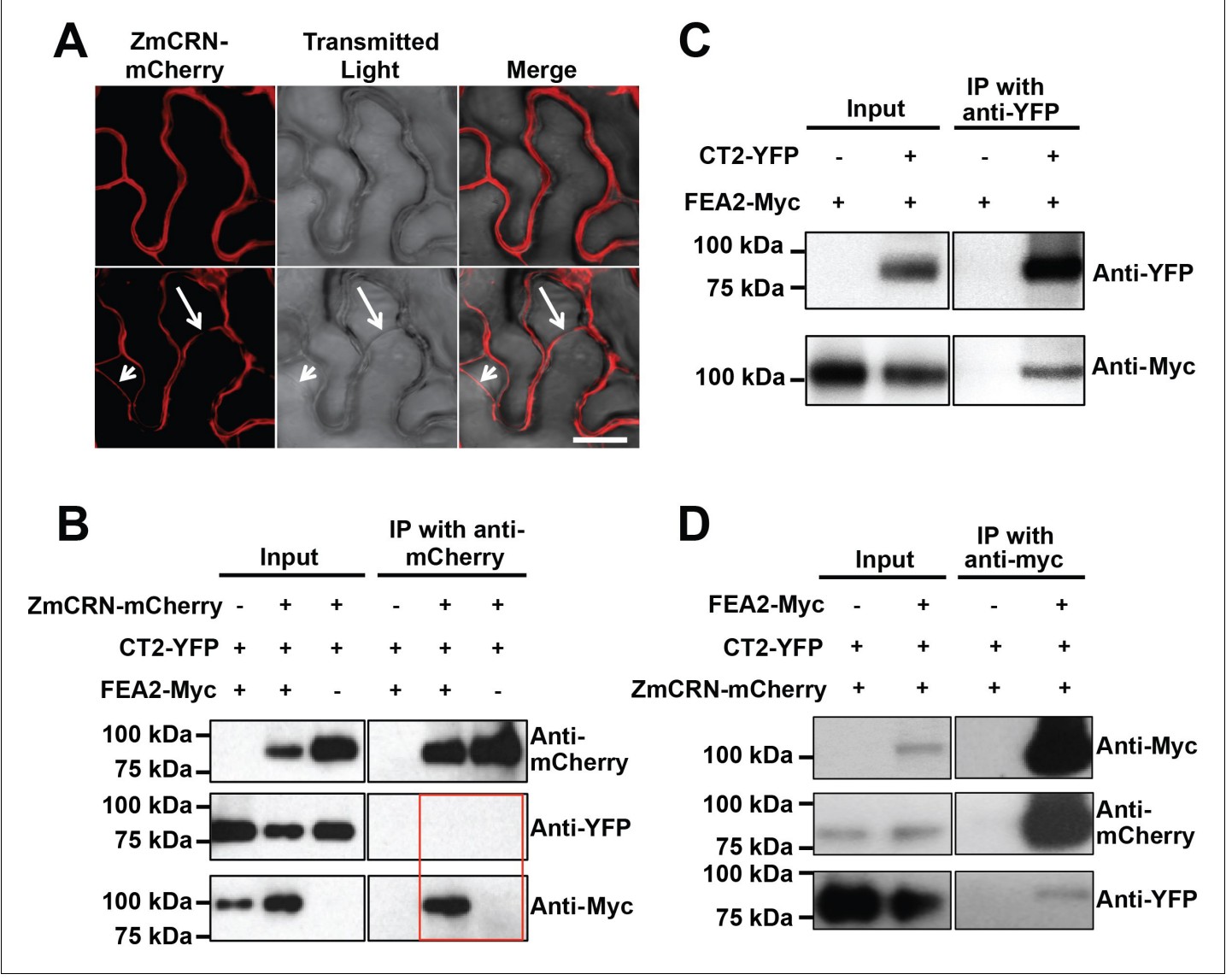

**Figure 4.** FEA2 is present in two different complexes. (A) ZmCRN-mCherry was localized at the plasma membrane following tobacco transient expression (top), and in subsequent plasmolysis (bottom). In transient expression followed by immunoprecipitation (IP) assay, ZmCRN-mCherry could IP FEA2-Myc, but not CT2-YFP (B), however CT2-YFP was able to IP FEA2-Myc, as expected (C). FEA2-Myc could also IP ZmCRN-mCherry and CT2-YFP, respectively (D). Scale bar: 20 μm in A.

DOI: https://doi.org/10.7554/eLife.35673.014

The following figure supplements are available for figure 4:

**Figure supplement 1.** ZmCRN-mCherry co-localized with FEA2-YFP and CT2-YFP on the plasma membrane.
DOI: https://doi.org/10.7554/eLife.35673.015

**Figure supplement 2.** FEA2-NmVen210 interacts with ZmCRN-CmVen210 by BiFC.
DOI: https://doi.org/10.7554/eLife.35673.016

**Figure supplement 3.** Treatment with ZmFCP1 or ZmCLE7 peptide didn't affect the protein complex formation.
DOI: https://doi.org/10.7554/eLife.35673.017

multiple environments (*Li et al., 2013*; *Liu et al., 2015*), suggesting that *ZmCRN* contributes to quantitative variation in this trait. Therefore, ZmCRN could be manipulated for maize yield enhancement.

Previously, we identified the alpha subunit of the heterotrimeric G protein, CT2, as an additional interactor of FEA2. *fea2* is epistatic to *ct2* in meristem regulation, similar to its genetic interaction

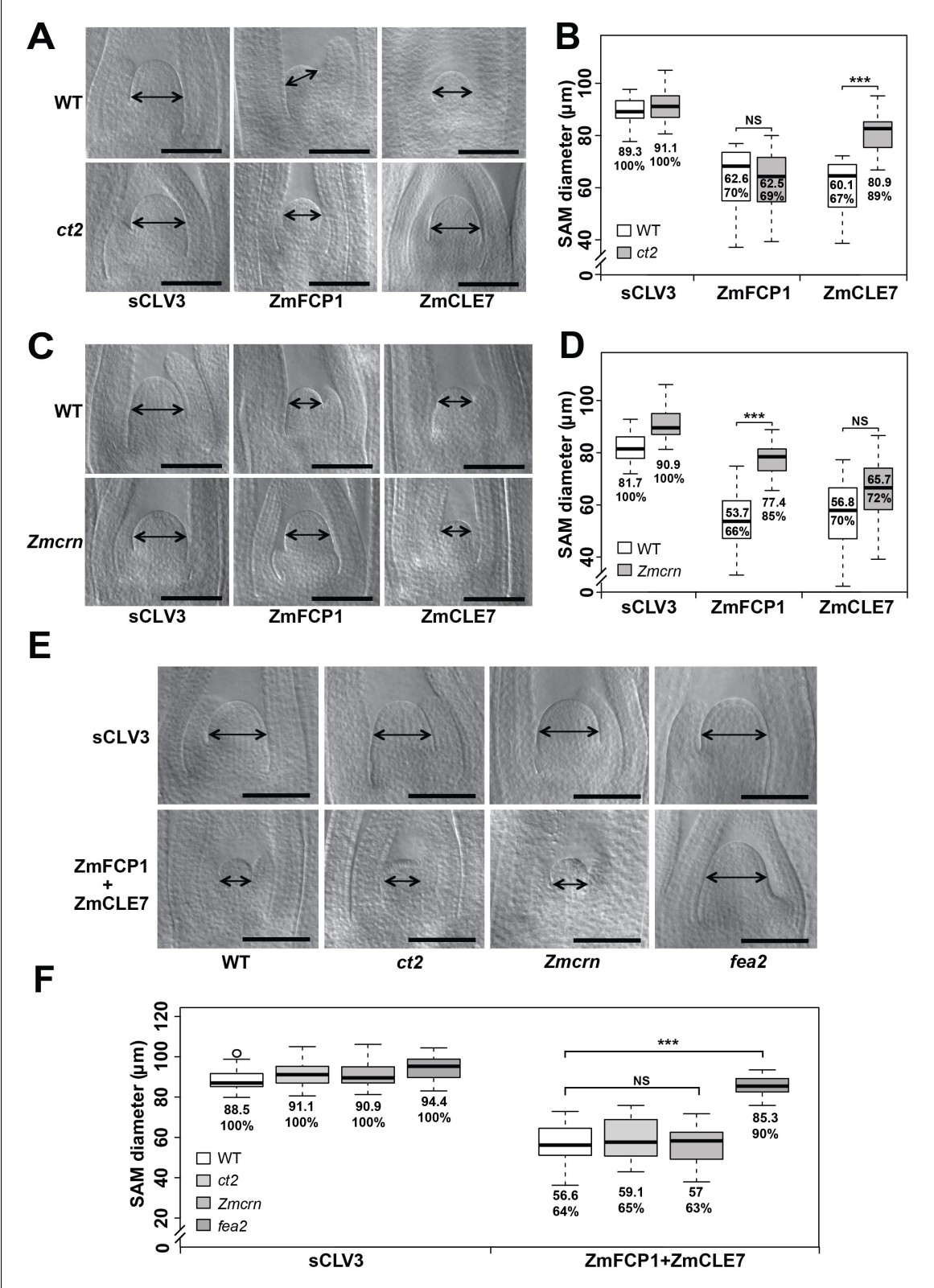

**Figure 5.** *ct2* and *Zmcrn* show different sensitivity to ZmCLE7 and ZmFCP1 peptides. Embryos of each genotype were cultured with control, scrambled peptide (sCLV3) or with ZmFCP1 or ZmCLE7. Wild type SAM growth (double-headed arrows) was strongly inhibited by all peptides except sCLV3, and *ct2* growth was insensitive only to ZmCLE7 peptide (A), whereas *Zmcrn* was partially resistant only to ZmFCP1 peptide (C); SAM diameter was quantified (B, D). In treatments with both ZmFCP1 and ZmCLE7, only *fea2* showed resistance, but *Zmcrn* or *ct2* did not (E, F). Scale bars: 100 μm in A, C

*Figure 5 continued on next page*

*Figure 5 continued*

and E. N = 25 (C) plants for each genotype. Data in B, D and F are shown by box plots. The mean values as well as the relative % to each negative control are listed. '***': P value < 0.0001, two-tailed, two-sample t test, 'NS': not significant.

DOI: https://doi.org/10.7554/eLife.35673.018

The following source data and figure supplements are available for figure 5:

**Source data 1.** ZmFCP1 and ZmCLE7 double peptide treatments in ct2,Zmcrnandfea2in a segregating population.

DOI: https://doi.org/10.7554/eLife.35673.023

**Figure supplement 1.** *ZmCRN* and *CT2* function in a different pathway to *FEA3*.

DOI: https://doi.org/10.7554/eLife.35673.019

**Figure supplement 1—source data 1.** SAM size of *Zmcrn;fea3* in a segregating population.

DOI: https://doi.org/10.7554/eLife.35673.020

**Figure supplement 2.** Triarabinosylated ZmCLE7 peptide is more potent.

DOI: https://doi.org/10.7554/eLife.35673.021

**Figure supplement 2—source data 1.** ZmCLE7-Arabinosylated peptide embryo assay.

DOI: https://doi.org/10.7554/eLife.35673.022

with *Zmcrn*, and FEA2 interacts with CT2 *in vivo*, revealing that *CT2*, like *ZmCRN*, is a candidate downstream signaling component of *FEA2* (*Bommert et al., 2013a*). Although *fea2* was epistatic both to *ct2* and to *Zmcrn*, we found that *ct2; Zmcrn* double mutants had an additive interaction, suggesting they function in parallel, and that the FEA2 signaling pathway branches into these two different downstream signaling components. This idea was supported by peptide assays in different mutants, which suggested that ZmCRN and CT2 function specifically in ZmFCP1 or ZmCLE7 signaling, respectively, while FEA2 is involved in both. Although we used high peptide concentrations, the activity of CLE peptides is known to be enhanced by triarabinosylation (*Ohyama et al., 2009*; *Matsubayashi, 2011*; *Xu et al., 2015*; *Corcilius et al., 2017*), and indeed we found that similarly modified ZmCLE7 peptide was about 10 fold more potent than the non-modified form (*Figure 5—figure supplement 2*).

Consistently with our findings, *ZmCRN*, *CT2* and *FEA2* were expressed broadly in the SAM in overlapping domains (*Figure 2—figure supplement 5*). These data suggest a novel mechanism in plant receptor signaling, where a single receptor, FEA2, can transmit signals from two different CLE peptides, ZmFCP1 and ZmCLE7, through two different downstream components, ZmCRN and CT2. We thereby shed light on how distinct signaling by different peptides can be achieved through a common receptor. As a candidate receptor or co-receptor for different peptides, FEA2 does not have any close homologs in the maize genome (*Figure 6—figure supplement 1*), similar to CLV2 in Arabidopsis, and the relatively mild phenotype of *fea2* mutants may be due to compensation by partially redundant parallel signaling pathways, such as through FEA3 (*Je et al., 2016*). Our results are largely consistent with findings in Arabidopsis, that CRN is dispensable for CLV3 perception and signaling (*Nimchuk, 2017*), and that CLV2/CRN can function with other CLE ligand-receptor complexes (*Hazak et al., 2017*). However, in Arabidopsis CRN is required for CLV2 trafficking to the plasma membrane (*Bleckmann et al., 2010*). Our results suggest that the maize CLV2 ortholog FEA2 still functions (with CT2) in a *crn* mutant, so is presumably on the plasma membrane even in the absence of ZmCRN.

How then can a single receptor recognize different signals and transmit them differentially? The most obvious answer depends on the hypothesis that FEA2 and CLV2 are co-receptors that function with LRR RLKs, which binds CLE peptides directly (*Figure 6*). This hypothesis is supported by the finding that CLV1 binds CLV3 with high affinity, but CLV2 is unable to bind CLE peptides (*Shinohara and Matsubayashi, 2015*), and that CLV2/CRN can function with different CLE ligand-receptor complexes (*Hazak et al., 2017*). There are conflicting results surrounding the interaction between CLV2 and CLV1; some experiments detect their physical interaction, but many of them use over-expression and are prone to false positive results, and *clv2* and *clv1* act additively in double mutant combinations (*Kayes and Clark, 1998*; *Müller et al., 2008*). This genetic result suggests they act separately, and the same is true for the orthologs *FEA2* and *TD1* in maize (*Bommert et al., 2005*). A possible explanation for these conflicting findings is that CLV2 may act with multiple CLE receptor RLKs. This model is supported by the observation that CLV1 homologs, the BAMs, function redundantly with CLV1, so multiple LRR RLKs do indeed function in meristem size control. This also

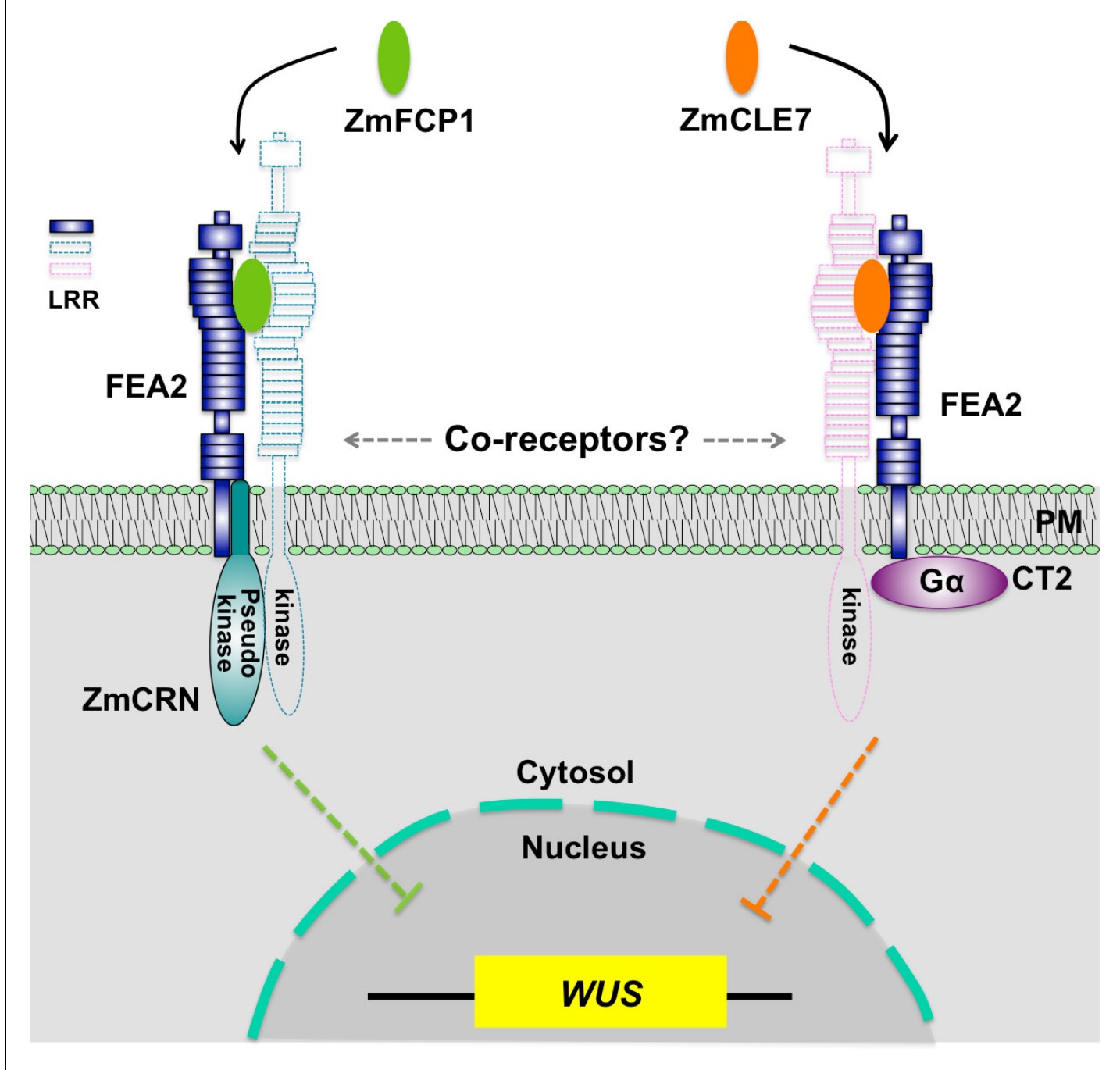

**Figure 6.** Hypothetical model for FEA2 signaling through two different pathways. Two different peptides, ZmFCP1 and ZmCLE7, are proposed to bind to two separate FEA2 receptor complexes, and the two signals are differentially transmitted to downstream components; with the ZmCLE7 signal passing through CT2, and the ZmFCP1 signal passing through ZmCRN.

DOI: https://doi.org/10.7554/eLife.35673.024

The following figure supplement is available for figure 6:

**Figure supplement 1.** Clustering, phylogenetic tree, and functional annotations of CLV2/FEA2 genes and their close homologs from five species.

DOI: https://doi.org/10.7554/eLife.35673.025

explains why all intermediate and strong *clv1* alleles are dominant negative, as they likely interfere with the activity of other receptor kinase(s) that have functional overlap with *CLV1* (*Diévart et al., 2003*; *Nimchuk et al., 2015*).

Despite not knowing the details of specific CLE-receptor interactions, our data show that FEA2 can transmit different peptide signals through two distinct downstream signaling components that most likely converge on the regulation of *ZmWUS* expression to regulate stem cell proliferation in meristem development (*Figure 6*). This suggests a new working model for meristem size regulation, in which ligand binding can be transmitted by a common co-receptor working with different RLKs coupled to distinct signaling proteins. Our model differs from most well-studied ligand-receptor signaling pathways, in which the signaling pathways usually converge (*Couto and Zipfel, 2016*). For instance, different microbial ligands such as flagellin and Elongation Factor Thermo unstable (EF-Tu) are specifically recognized by the FLAGELLIN-SENSITIVE 2 (FLS2)-BRI1 ASSOCIATED RECEPTOR KINASE (BAK1) or EF-Tu RECEPTOR (EFR)-BAK1 RLK complexes, respectively, while signal transduction requires a shared set of cytosolic kinases, including BOTRYTIS-INDUCED KINASE 1 (BIK1) (*Aarts et al., 1998*; *Lu et al., 2010*). Nevertheless, a similar principle can be drawn from the different signaling pathways mediated by BAK1, which functions as a co-receptor for the brassinosteroid (BR) receptor, BR INSENSITIVE 1 (BRI1) or for FLS2. After ligand perception, BR signaling through the BAK1-BRI1 complex is transmitted through the receptor-like cytoplasmic kinase (RLCK) BRASSI NOSTEROID-SIGNALING KINASE 1 (BSK1), and flagellin signaling through the BAK1-FLS2 complex is transmitted through a different RLCK, BIK1 (*Li et al., 2002*; *Nam and Li, 2002*; *Chinchilla et al., 2007*; *Lu et al., 2010*; *Wang, 2012*; *Sun et al., 2013*). Our study also reveals another source of variation in meristem receptor signaling, by highlighting the role of an additional CLE peptide, ZmFCP1. The role of FCP1 in meristem maintenance has been characterized in both maize and rice (*Suzaki et al., 2008*; *Je et al., 2016*), but not yet in *Arabidopsis*.

In summary, multiple receptor signaling pathways appear to be required to for the perception of different CLE peptide signals to fine tune meristem development. This complex system of multiple peptides, receptors and downstream components presumably confers robustness on the meristem structure, as well as providing flexibility to control meristem development according to different physiological or developmental cues. For example, meristem size responds to stress and developmental transitions, such as floral induction, and different signaling pathways may confer such responsiveness. Our results help explain how meristem size regulation is orchestrated by multiple CLE peptides and receptors, as observed in many species including Arabidopsis, rice, maize and tomato (*Ito et al., 2006*; *Strabala et al., 2006*; *Suzaki et al., 2009*; *Nimchuk et al., 2015*; *Xu et al., 2015*). They also support the idea that meristem signaling components are highly conserved between diverse plant species, and a major challenge is to understand how differential regulation of these common components leads to diversity in meristem organization and size across diverse plant taxa.

## Materials and methods

### Plant growth and map based cloning

Maize plants were grown in the field or in the greenhouse. The *Zmcrn Mu* insertion allele was isolated from TUSC lines and was backcrossed three generations to the standard B73 inbred line. The *fea*148* allele was isolated in an EMS mutagenesis screen using F2 seed stocks prepared by Prof. Gerald Neuffer, derived from a cross of mutagenized B73 pollen onto A619 ears. One fasciated plant from the segregating *fea*148* M2 population from the maize GDB stock center was crossed to the A619 inbred, then selfed to make an F2 segregating population. Pooled DNAs from ~50 mutants or the same number of normal ear plants screened from the segregating F2 population were used for bulked segregant analysis (BSA) using a maize SNP50 chip (Illumina, Inc.). The BSA analysis revealed a clear linkage of the mutation on Chromosome 3 at 153–158 Mbp. As *ZmCRN* was an obvious candidate gene within the region, we sequenced the locus of *ZmCRN* using the mutant pool DNA and found a C to T mutation in the pseudokinase domain, which led to an early stop codon.

To measure meristem size, segregating siblings were genotyped and shoot apices of 7-day-old plants (*Figure 2B*) or 21-day-old plants (*Figure 3A and D*) were dissected, cleared and measured as described previously (*Taguchi-Shiobara et al., 2001*). Measurement was made blindly without the

knowledge of the genotypes. All measurements included at least 10 samples of each genotype, and two or three independent biological replicates, and mean values ± s.d. were presented, with significance calculated using two-tailed, two-sample $t$ tests, and significant differences reported as $P$ values.

## Imaging

Scanning electron microscopy was performed on fresh tissues of maize using a Hitachi S-3500N SEM, as described (*Taguchi-Shiobara et al., 2001*). For confocal microscopy, tobacco infiltrated tissues were dissected and images were taken with a Zeiss LSM 710 microscope, using 561 nm laser excitation and 580–675 nm emission for detection ZmCRN-mCherry, using 512 nm laser excitation and 518–538 nm emission for detection of CT2-YFP and FEA2-YFP and for BiFC imaging. For plasmolysis of ZmCRN-mCherry, leaf tissues were incubated for 30 min with 800 mM mannitol and imaged.

## Double mutant analysis and in situ hybridization

Double mutants were constructed by crossing mutants introgressed into B73, followed by selfing or backcrossing to the F1. All plants were subsequently genotyped (primers are listed in Supplementary file 2). In situ hybridization experiments were performed as described (*Jackson et al., 1994*). Antisense and sense RNAs for *ZmCRN* were transcribed and used as probes. Primers are listed in *Supplementary file 2*.

## Protein expression and Co-IP assays

*CT2-YFP*, *ZmCRN-mCherry*, or *FEA2-Myc* expression constructs were infiltrated into 4-week-old *Nicotiana benthamiana* leaves together with a P19 plasmids to suppress posttranscriptional silencing (*Mohammadzadeh et al., 2016*). The protein extraction and membrane fraction enrichment were described in *Bommert et al., 2013a*. Briefly, the infiltrated leaves were harvested 3-d post infiltration. The leaf tissues were ground in liquid nitrogen to a fine powder then suspended in twice the volume of protein extraction buffer containing 150 mM NaCl, 50 mM Tris-HCl pH 7.6, 5% glycerol, and EDTA-free Protease inhibitor cocktail (Roche). After filtration through Miracloth, and centrifugation at 4,000 g for 10 min at 4°C, the extract was centrifuged at 100,000 g for 1 hr at 4°C to enrich the microsomal membrane fraction. The resulting pellet was re-suspended in the extraction buffer supplemented with 1% Triton X-100. Lysates were cleared by centrifugation at 100,000 g for 30 min at 4°C to remove non-solubilized material. ZmCRN-mCherry was immunoprecipitated using RFP-Trap (Chromotek) in membrane solubilization buffer for 40 min followed by washing 3 times with 1 ml of the same buffer. The IP'd proteins were eluted with 50 µl 1xSDS loading buffer at 95°C, followed by standard SDS-PAGE electrophoresis and western blotting. FEA2-Myc was immunoprecipitated using agarose beads conjugated with anti-Myc antibody (Millipore, 16–219, RRID:AB_390197). ZmCRN-mCherry was detected using an anti-RFP antibody (Rockland, 600-401-379, RRID:AB_2209751), FEA2-Myc was detected using an anti-Myc antibody (Millipore, 05–724, RRID:AB_309938), and CT2-YFP was detected using an anti-GFP antibody (Roche, 11814460001, RRID:AB_390913).

## Peptide assays

Maize embryos segregating for each mutant were dissected at 10 days after pollination, when the SAM was exposed, and cultured on gel media (*Bommert et al., 2013a*) containing scrambled peptide (30 µM; Genscript) or ZmFCP1 peptide or ZmCLE7 peptide or a mixture of ZmCLE7 and ZmFCP1 peptides (*Je et al., 2016*). After 12 days, the tissues were harvested for genotyping and the embryos were fixed in FAA (10%, formalin, 5% acetic acid, 45% ethanol) and cleared in methyl salicylate, and SAMs measured by microscopy, as described (*Je et al., 2016*). Triarabinosylated peptides were synthesized as described (*Corcilius et al., 2017*).

## Two-components transactivation assay

The two-component transactivation assay was performed as described (*Je et al., 2016*), and the lines were backcrossed into the *fea3* mutant background. To measure meristem size, segregating siblings were genotyped and shoot apical meristems of 14-day-old plants (*Figure 1A*) were dissected, cleared and measured as described previously (*Taguchi-Shiobara et al., 2001*).

## Association analysis of the *ZmCRN* locus

The candidate gene association analysis of *ZmCRN* with the kernel row number (KRN) trait was conducted in a maize association panel with 368 diverse inbred lines (*Li et al., 2013*). 22 SNPs in the *ZmCRN* gene region were observed based on previously released genotypes in the association panel. This was combined with KRN phenotypic data from five environments and BLUP (Best Linear Unbiased Prediction) data, including in Ya'an (30°N, 103°E), Sanya (18°N, 109°E) and Kunming (25°N, 102°E) in 2009 and Wuhan (30°N, 114°E) and Kunming (25°N, 102°E) in 2010 (*Liu et al., 2015*). The association between *ZmCRN* and KRN was established by a mixed linear model corrected by population structure, with p-value<0.001 as threshold (*Zhang et al., 2010*; *Li et al., 2013*).

## Phylogenetic analysis

CLAVATA2 and FASCIATED EAR2 orthologs from *Arabidopsis thaliana*, *Solanum lycopersicum*, *Zea mays*, *Oryza sativa*, and *Amborella trichopoda* were aligned using MUSCLE (*Edgar, 2004*; *Ouyang et al., 2007*; *Lamesch et al., 2012*; *Tomato Genome Consortium, 2012*; *Amborella Genome Project, 2013*; *Jiao et al., 2017*). This alignment was converted to a Hidden Markov Model (HMM) using HMMER3.1b2 (hmmer.org), and was used to identify sequences that bore homology within the genomes of these five species (e-value cutoff <10e-3). These amino acid sequences were grouped using convex clustering in CLANS (*Frickey and Lupas, 2004*), and sequences that did not cluster closely with the CLV2/FEA2 cluster were removed manually, followed by subsequent clustering; this was repeated until no sequences were identified as separate from the CLV2/FEA2 cluster. Initial phylogenetic analyses of these sequences revealed a clade of RLPs sister to the CLV2/FEA2 clade. This subset of RLP sequences was used to build two additional HMMs as described above (hmmer.org), one of which included only monocot RLP sequences. These two RLP HMMs were used to search the five focal genomes again. All of the sequences recovered using both RLP HMMs were combined with the refined subset identified with the CLV2/FEA2 HMM, and iteratively clustered using CLANS until no sequences were identified as separate from the CLV2/FEA2 cluster (*Frickey and Lupas, 2004*). The final set of sequences, with any kinase domains removed, were aligned via MAFFT L-INS-I (*Katoh et al., 2005*; *Katoh and Standley, 2013*). Model selection was performed using PartitionFinder2 (*Lanfear et al., 2017*) and phylogenetic analysis under the maximum likelihood information criterion was performed using RAxML with the VT + I + G model and 1000 bootstrap replicates (*Stamatakis, 2014*). Signal peptide and transmembrane domains were identified using Phobius, and the presence of a kinase domain was determined using HMMER3.1b2 and the Pkinase domain, respectively (*Käll et al., 2004*; *Finn et al., 2016*; hmmer.org).

## Acknowledgements

We thank Prof. Gerald Neuffer, and the Maize Genetics Stock Center for the *fea*148* mutant, Tim Mulligan and Sarah Vermylen for plant care, Tara Skopelitis for assistance in genotyping, Prof. Zachary Lippman, Prof. Peter Bommert and members of the Jackson lab for comments on the manuscript, and acknowledge funding from the Agriculture and Food Research grant no. 2015–06319 and 2013–02198 of the USDA National Institute of Food and Agriculture, a collaborative agreement with Dupont Pioneer, and from NSF Plant Genome Research Program grants # IOS-1238202, IOS-1546837 and MCB-1027445, 'Next-Generation BioGreen 21 Program (SSAC, Project No. PJ01184302 and PJ01322602), Rural Development Administration, Republic of Korea, and HFSP Long-Term fellowship program (LT000227/2016-L).

## Additional information

### Funding

| Funder | Grant reference number | Author |
| --- | --- | --- |
| National Institute of Food and Agriculture | 2015-06319 | David Jackson |
| National Science Foundation | IOS-1238202 | David Jackson |

| Next-Generation BioGreen 21 Program | PJ01184302 | David Jackson |
|---|---|---|
| Human Frontier Science Program | Postdoctoral Fellowships, LT000227/2016-L | Fang Xu |
| National Institute of Food and Agriculture | 2013-02198 | David Jackson |
| National Science Foundation | MCB-1027445 | David Jackson |
| National Science Foundation | IOS-1546837 | Madelaine E Bartlett David Jackson |
| Next-Generation BioGreen 21 Program | PJ01322602 | David Jackson |

The funders had no role in study design, data collection and interpretation, or the decision to submit the work for publication.

### Author contributions

Byoung Il Je, Conceptualization, Data curation, Formal analysis, Investigation, Methodology, Writing—original draft, Writing—review and editing; Fang Xu, Conceptualization, Data curation, Formal analysis, Funding acquisition, Investigation, Methodology, Writing—review and editing; Qingyu Wu, Lei Liu, Data curation, Methodology; Robert Meeley, Conceptualization, Formal analysis, Methodology; Joseph P Gallagher, Leo Corcilius, Richard J Payne, Madelaine E Bartlett, Conceptualization, Methodology; David Jackson, Conceptualization, Data curation, Formal analysis, Supervision, Funding acquisition, Investigation, Methodology, Writing—original draft, Writing—review and editing

### Author ORCIDs

Byoung Il Je https://orcid.org/0000-0002-6661-5855
Fang Xu https://orcid.org/0000-0003-0767-1272
Qingyu Wu https://orcid.org/0000-0003-3064-2445
Robert Meeley https://orcid.org/0000-0002-4496-1888
Richard J Payne https://orcid.org/0000-0002-3618-9226
David Jackson https://orcid.org/0000-0002-4269-7649

### Decision letter and Author response

Decision letter https://doi.org/10.7554/eLife.35673.030
Author response https://doi.org/10.7554/eLife.35673.031

## Additional files

### Supplementary files

• Supplementary file 1. table 1. *P*-values of the association between ZmCRN SNPs with kernel row number in multiple environments.
DOI: https://doi.org/10.7554/eLife.35673.026

• Supplementary file 2. table 2. List of Primers.
DOI: https://doi.org/10.7554/eLife.35673.027

• Transparent reporting form
DOI: https://doi.org/10.7554/eLife.35673.028

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
