## [Decision Letter]

[Editors’ note: a previous version of this study was rejected after peer review, but the authors submitted for reconsideration. The first decision letter after peer review is shown below.]

Thank you for submitting your work entitled "The CLAVATA receptor FASCIATED EAR2 responds to distinct CLE peptides by signaling through two downstream effectors" for consideration by *eLife*. Your article has been reviewed by three peer reviewers, and the evaluation has been overseen by a Senior/Reviewing Editor.

Our decision has been reached after consultation between the reviewers. Based on these discussions and the individual reviews below, we regret to inform you that your work cannot be considered for publication in *eLife* without substantial improvements that go much beyond a simple round of revisions.

The reviewers all agree that your manuscript presents an interesting story. The concept emerging from your work is potentially exciting and important, however, in the opinion of the reviewers and editors, it is not yet supported rigorously by the data. To make the strong point you project here, a number of additional, crucial experiments would be required. This includes, for example, expression pattern analyses, double mutant analyses, independent protein-protein interaction verifications, or more realistic physiological conditions, like lower peptide concentrations. If you choose to substantially improve the manuscript along these lines, and comprehensively address the individual reviewer comments pasted below, we are however ready to take another look at a resubmitted manuscript.

*Reviewer #1:*

An unresolved mystery in plant development concerns the developmental mechanisms whereby functional specificity can arise from the redundancy of receptors involved in controlling stem cell homeostasis in shoot meristems. This well written manuscript addresses this question, to describe how the maize CLV2 gene can interact with two distinct peptide ligands by associating with two distinct effectors, which presumably in different downstream signaling pathways. This makes a novel contribution to the field.

Introduction.

1) The introduction should stipulate that no WUS function is yet demonstrated in maize, and that the CLV1 homolog does not have a demonstrated function in the maize SAM. The manuscript says: "In maize, THICK TASSEL DWARF1 (TD1) and FEA2 are CLV1 and CLV2 orthologs, and function similarly to restrict SAM proliferation" – this is untrue – TD1 mutations affect the inflorescence shoot meristems, but there is no evidence that TD1 restricts vegetative SAM size.

Results.

2) Why is there no demonstration of ZmCRN RNA or protein accumulation in maize shoot meristems?

3) A new maize mutation in ZmCRN is identified and characterized. The genetic data clearly demonstrate that FEA2/CLV2 is epistatic to both ZmCRN and CT2, and that ZmCRN and CT2 function in different pathways.

4) Although the CT2 YFP fusion was shown previously to complement the mutant phenotype, no assays of FEA2 or CRN fusion protein function are provided. Although the data, derived from a heterologous system, are clear and indicate the FEA2/CLV2 can interact with both CT2 and ZmCRN – while CT2 and ZmCRN do not form a complex in tobacco – I wonder why these fusion proteins were not introduced into maize? This is not always trivial due to complex promoter issues, but should be explained.

5) The data clearly show that FEA2 mutants are reduced in sensitivity to both the FCP and CLE7 peptides, CT2 shows reduced sensitivity to its presumptive ligand CLE7 (but not FCP1), and ZmCRN is reduced in sensitivity to FCP1 (its presumptive ligand), but is sensitive to CLE7. However, the legend to Figure 5 describes *crn* mutants as insensitive to FCP1, whereas the text better describes this interaction as "partially resistant", since FEA3 is presumably intact and should be responding to applied FCP ligand. Best to fix this inconsistency between the descriptions in the legend and the manuscript text.

*Reviewer #2:*

The manuscript by Je et al. reports genetic and phenotypic characterizations among the mutants of CLAVATA pathway in Maize. The authors isolated a loss-of-function allele of ZmCRN, and characterized its meristem phenotype with *fea2*, that lacks a functional CLV2 ortholog, *fea3*, an LRR-RLK previously reported by the same group, and *ct2*, a mutant in G-α. Together with a series of double mutant analysis, CLE peptide application experiments, and in plant Co-IP experiments in *N. benthamiana*, the authors conclude that FEA2 form a different receptor complex depending on the CLE peptide.

The manuscript brings an important insight into a key question in receptor signaling, "how could a receptor recognizes and discriminates related yet different signals to activate specific response?". At the same time, however, the data provided are rather slim, and this reviewer feels that such a major claim should come with solid and multi-faceted experimental data of high rigor. Specific comments are as the following:

Role of FEA3: The genetic interactions of FEA3 in FEA2-CT and FEA2-ZmCRN pathways needs to be addressed in order to evaluate the genetic interaction and peptide application data. Both *fea2* and *fea3* are resistant to ZmFCP1 peptide application (Figure 1). Because of this, partial sensitivity of Zmcrn mutants to ZmFCP1 (Figure 5) may be mediated through FEA3. Peptide application on *fea3* Zmcrn double mutants would clarify this.

ZmCRN expression/localization patterns: Where is ZmCRN expressed? For ZmCRN to function via forming a heterodimer with FEA2, their expression domain should overlap within the SAM. The authors could generate transgenic lines expressing the FP-tagged CRN in maize (which I agree might be very difficult and takes time for maize), or at least the authors must show clear in situ hybridization data long with the FEA2 and CT2.

Subcellular co-localization in *N. benthamiana*: The authors show that ZmCRN-mCherry localizes to PM in tobacco leaf epidermis. Would ZmFEA2 and/or CT co-localizes with ZmCRN on the plasma membrane in tobacco?

Protein-protein interaction assays: The authors present only one type experiment (Co-IP) to conclude that FEA2 forms a separate complex with ZmCRN and CT2 depending on the ligands. The following experiments would strengthen the conclusion:

a) Reciprocal Co-IP. The authors should reciprocally IP FEA2 and confirm that FEA2 indeed brings down BOTH CT2 and ZmCRN when three proteins are co-expressed. I think this experiment must be performed.

b) Role of different peptides, ZmFCP1 and ZmCLE7, on receptor complex formation. This can be done by treating/infiltrating/co-expressing each peptide to the *N. benthamiana* leaves along with a pair-wise combination of receptors.

Alternative experimental approach to support the in planta Co-IP experiments. The authors could perform BiFC, split-Ub or other protein-protein interaction assays to strengthen their claim.

Figure 4. The gel lanes appear spliced. This is not an accepted practice or data presentation.

Discussion, fourth paragraph (discussion about why *clv2* phenotype is weaker than *clv1*). A previous report (Diévart et al., 2003) shows that missense alleles of *clv1* act in a dominant-negative manner. Please include this into the Discussion.

*Reviewer #3:*

The manuscript by Je et al. reports the reverse (and forward) genetic characterization of ZmCRN, the mays ortholog of CORYNE (CRN) in *Arabidopsis*. The authors provide compelling genetic evidence that, just like in *Arabidopsis*, CRN acts in a common pathway with the LRR receptor kinase CLV2 (FEA2 in Zm). The novel finding is that FEA2 can form complexes with either ZmCRN or the heterotrimeric G-protein α subunit ZmCT2. The phenotypes of ZmCRN and ZmCT2 are additive, suggesting that FEA2 can signal via two different down-stream components. Peptide resistance assays furthermore suggest that these different signaling complexes may form in response to the sensing of different CLE peptides, ZmCLE7 and ZmFCP1, respectively. Overall this is a well written manuscript which provides some novel insight into the CLE signaling pathway involved in plant stem cell maintenance. Since, I am not a maize geneticist, I will focus my review on a) the biochemical experiments presented in this study and b) on the discussion of these novel findings:

a) I understand that it is very difficult to perform any co-IP experiments in maize and that for this reason a transient system was chosen. I would find it nevertheless appropriate to at least mention in the Discussion that the outlined interactions were assayed not with native-like protein levels but rather by strongly overexpressing CRN, FEA2 and CT2.

b) Why are the peptide resistance assays done with such high peptide concentrations (30 μM)? Do the authors think that the binding affinities of the maize receptor complexes are much lower than in *Arabidopsis*? Or are there post-translational modifications missing from the synthetic peptides? Or are the peptides poorly taken up by the plant tissue?

c) Based on their findings, the authors suggest FEA2/CLV2 as a co-receptor for several CLE peptide-sensing LRR receptor kinases. In terms of the presented as well as previously established genetics on FEA2/CLV2 this would only make sense, if indeed CLV2 function can be replaced by another co-receptor in vivo. I would appreciate if the authors could present here a sequence alignment of the known receptor-like proteins from maize. Are there sequence homologs for CLV2 in maize (there are no obvious homologs in *Arabidopsis*)?

---

## [Author Response]

[Editors’ note: the author responses to the first round of peer review follow.]

The reviewers all agree that your manuscript presents an interesting story. The concept emerging from your work is potentially exciting and important, however, in the opinion of the reviewers and editors, it is not yet supported rigorously by the data. To make the strong point you project here, a number of additional, crucial experiments would be required. This includes, for example, expression pattern analyses, double mutant analyses, independent protein-protein interaction verifications, or more realistic physiological conditions, like lower peptide concentrations. If you choose to substantially improve the manuscript along these lines, and comprehensively address the individual reviewer comments pasted below, we are however ready to take another look at a resubmitted manuscript.

We appreciate the constructive criticisms and suggestions from the three expert reviewers. We have added new data to address all comments, including expression analysis of *ZmCRN, fea3;Zmcrn* double mutant analyses, independent protein-protein interaction verification and treatments with different peptide concentrations, to address the issues. This new data further supports our major conclusion, and the manuscript has been substantially revised accordingly.

Reviewer #1:[…] Introduction.1) The introduction should stipulate that no WUS function is yet demonstrated in maize, and that the CLV1 homolog does not have a demonstrated function in the maize SAM. The manuscript says: "In maize, THICK TASSEL DWARF1 (TD1) and FEA2 are CLV1 and CLV2 orthologs, and function similarly to restrict SAM proliferation" – this is untrue – TD1 mutations affect the inflorescence shoot meristems, but there is no evidence that TD1 restricts vegetative SAM size.

As suggested, we have added text to clarify that WUS has not been functionally characterized in maize, and also added information about rice WUS function:

“Two maize WUS orthologs, ZmWUS1 and ZmWUS2, were predicted by phylogenetic analysis, and a *ZmWUS1* reporter is expressed in the presumptive organizing center of the inflorescence shoot meristem (Je at al., 2016), but these genes have not been functionally characterized (Nardmann and Werr 2006). […] The rice WUS homolog, *TILLERS ABSENT1/ MONOCULM3* functions in axillary shoot meristem formation (Tanaka et al., 2015; Lu et al., 2015), and WUS function in the shoot apical meristems appears to have been taken over by the *WUSCHEL RELATED HOMEOBOX4 (WOX4)* gene (Ohmori et al., 2013)”

As to whether these mutations affect SAM or inflorescence shoot meristems, this is a semantic point, since the inflorescence meristem is also a “SAM”, as it is the shoot meristem at the apex of the inflorescence. In general, the fasciated mutations in maize do enlarge the vegetative SAM, but in a much less dramatic way. However, to clarify, we changed “SAM” to “inflorescence shoot meristem” in the Introduction (“… and function similarly to restrict inflorescence shoot meristem proliferation…”).

Results.2) Why is there no demonstration of ZmCRN RNA or protein accumulation in maize shoot meristems?

In the revised manuscript, we now show expression of *ZmCRN* byin situ hybridization in maize shoot meristems. *ZmCRN* was expressed throughout the SAM and more strongly in the peripheral domain and leaf primordia (Figure 2). This profile is confirmed by laser capture microdissection RNAseq data for different domains of the SAM (data from Timmermans lab, available from MaizeGDB; https://maizegdb.org/). This new data is added as Figure 2 and Figure 2—figure supplement 4, and new text:

“*ZmCRN* was expressed throughout the SAM and more strongly in the peripheral domain and leaf primordia (Figure 2, confirmed by laser capture microdissection RNAseq, Figure 2—figure supplement 4).”

3) A new maize mutation in ZmCRN is identified and characterized. The genetic data clearly demonstrate that FEA2/CLV2 is epistatic to both ZmCRN and CT2, and that ZmCRN and CT2 function in different pathways.

Thanks for the positive comment!

4) Although the CT2 YFP fusion was shown previously to complement the mutant phenotype, no assays of FEA2 or CRN fusion protein function are provided. Although the data, derived from a heterologous system, are clear and indicate the FEA2/CLV2 can interact with both CT2 and ZmCRN – while CT2 and ZmCRN do not form a complex in tobacco – I wonder why these fusion proteins were not introduced into maize? This is not always trivial due to complex promoter issues, but should be explained.

We indeed previously made pFEA2-FEA2-YFP transgenic maize plants, and saw clear FEA2-YFP signal on the plasma membrane in the inflorescence meristem (http://maize.jcvi.org/cellgenomics/geneDB_list.php. However, we didn’t make the ZmCRN transgenic plants, as the reviewer points out this is costly and time consuming. However, as requested by reviewer 2, we have added evidence of the plasma membrane localization of ZmCRN and additional support for protein-protein interactions, by performing reciprocal Co-IP and BiFC. These data were added in the revised manuscript (Figure 4, Figure supplement 6 and Figure supplement 7). These additional assays give us extra confidence in the plasma membrane localization of ZmCRN and interactions between ZmCRN and FEA2, even though the experiments were carried out in *N. benthamiana*. Please note that we do not detect FEA2-CT2 direct interaction using BiFC, even though we have confirmed their interaction many times by in vivo co-IP, and we believe another protein is bridging their interaction (Q. Wu and DJ, unpublished).

“FEA2-YFP, ZmCRN-mCherry, CT2-YFP and ZmCRN-mCherry were also co-localized on the plasma membrane when they were co-expressed (Figure 4—figure supplement 1). […] Lastly, as expected, no signal was detected when CT2-NmVen210 was co-expressed with ZmCRN-CmVen210 (Figure 4—figure supplement 2), confirming out co-IP results, and supporting the hypothesis that they do not interact.”

5) The data clearly show that FEA2 mutants are reduced in sensitivity to both the FCP and CLE7 peptides, CT2 shows reduced sensitivity to its presumptive ligand CLE7 (but not FCP1), and ZmCRN is reduced in sensitivity to FCP1 (its presumptive ligand), but is sensitive to CLE7. However, the legend to Figure 5 describes crn mutants as insensitive to FCP1, whereas the text better describes this interaction as "partially resistant", since FEA3 is presumably intact and should be responding to applied FCP ligand. Best to fix this inconsistency between the descriptions in the legend and the manuscript text.

Thanks for catching this inconsistency. We reworded the Figure 5 legend to “Zmcrn was partially resistant only to ZmFCP1 peptide” to make it consistent as suggested.

Reviewer #2:[…] Role of FEA3: The genetic interactions of FEA3 in FEA2-CT and FEA2-ZmCRN pathways needs to be addressed in order to evaluate the genetic interaction and peptide application data. Both fea2 and fea3 are resistant to ZmFCP1 peptide application (Figure 1). Because of this, partial sensitivity of Zmcrn mutants to ZmFCP1 (Figure 5) may be mediated through FEA3. Peptide application on fea3 Zmcrn double mutants would clarify this.

We agree with the reviewer that it is important to understand the interaction between *FEA3* and *ZmCRN*, since both factors act to transmit the ZmFCP1 signal. We therefore added new data on the genetic interactions between *Zmcrn* and *fea3* (and also *ct2* and *fea3*) in double mutants. In the segregating double mutant population, the *fea3; Zmcrn* double mutants SAMs were larger than the single mutants,suggesting that FEA3 and ZmCRN do not function in a common pathway. Similar findings were observed for *fea3; ct2* double mutants. Thus, *fea3* acts additively with *Zmcrn*, suggesting that ZmFCP1 signaling is mediated by two parallel pathways, one through FEA3 working through an as yet unknown downstream component(s), and another through FEA2 coupled with ZmCRN. This genetic data largely supports the reviewer’s speculation that the partial sensitivity of *Zmcrn* mutants to ZmFCP1 is mediated through FEA3. Importantly, our data suggest that *CT2* and *ZmCRN* do not act as downstream components for *FEA3*.

We have not performed peptide assays with the *fea3; Zmcrn* double mutants, since we just obtained them and it would take an additional 4-5 months to grow plants to make embryos and complete the treatments. We hope that the new data on *fea3; Zmcrn* double mutants is sufficient to convince the reviewer that *FEA3* and *ZmCRN* are acting in parallel, thus explaining the partial sensitivity of *Zmcrn* mutants to ZmFCP1 peptide.

These new data are presented in Figure 5—figure supplement 1,and in the text:

“As FEA3 also acts to transmit the ZmFCP1 signal (Je et al., 2016), we used genetic analysis to ask if ZmCRN also functions downstream of FEA3. […] Thus, ZmFCP1 signaling appears to be mediated by two different pathways, one acting through FEA2 coupled with ZmCRN, and another acting through FEA3 working through as yet unknown downstream component(s).”

ZmCRN expression/localization patterns: Where is ZmCRN expressed? For ZmCRN to function via forming a heterodimer with FEA2, their expression domain should overlap within the SAM. The authors could generate transgenic lines expressing the FP-tagged CRN in maize (which I agree might be very difficult and takes time for maize), or at least the authors must show clear in situ hybridization data long with the FEA2 and CT2.

Thanks for understanding the difficulty and long time to make FP-tagged CRN transgenic plants. As also requested by reviewer 1, we carried out in situ hybridization with maize shoot meristems using a *ZmCRN* anti-sense probe. *ZmCRN* was expressed throughout the SAM and more strongly in the peripheral domain and leaf primordia (Figure 2). This profile was confirmed by laser capture microdissection RNAseq data for different domains of the SAM (data from Timmermans lab, available from MaizeGDB; https://maizegdb.org/). This new data is added as Figure 2 and Figure 2—figure supplement 4. CT2 has been shown previously to express broadly in the SAM using CT2-YFP transgenic plants (Bommert et al., 2013a). In addition, laser capture RNA seq data showed that *FEA2, ZmCRN* and *CT2* are all expressed broadly in all domains of the SAM, suggesting that they are co-expressed (Figure 2—figure supplement 4).

This new data is added as Figure 2 and Figure 2—figure supplement 4, and new text:

*“ZmCRN* was expressed throughout the SAM and more strongly in the peripheral domain and leaf primordia (Figure 2, confirmed by laser capture microdissection RNAseq, Figure 2—figure supplement 4).”

Subcellular co-localization in N. benthamiana: The authors show that ZmCRN-mCherry localizes to PM in tobacco leaf epidermis. Would ZmFEA2 and/or CT co-localizes with ZmCRN on the plasma membrane in tobacco?

Yes, we observed that FEA2 were co-localized with ZmCRN on the plasma membrane when these two proteins were co-expressed in *N. benthamiana* (Figure supplement 6). We also saw co-localization of CT2-YFP with ZmCRN-mCherry on plasma membrane when they are co-expressed (Figure 4—figure supplement 1):

“FEA2-YFP, ZmCRN-mCherry, CT2-YFP and ZmCRN-mCherry were also co-localized on the plasma membrane when they were co-expressed (Figure 4—figure supplement 1).”

Protein-protein interaction assays: The authors present only one type experiment (Co-IP) to conclude that FEA2 forms a separate complex with ZmCRN and CT2 depending on the ligands. The following experiments would strengthen the conclusion:a) Reciprocal Co-IP. The authors should reciprocally IP FEA2 and confirm that FEA2 indeed brings down BOTH CT2 and ZmCRN when three proteins are co-expressed. I think this experiment must be performed.Alternative experimental approach to support the in planta Co-IP experiments. The authors could perform BiFC, split-Ub or other protein-protein interaction assays to strengthen their claim.

As suggested, we performed the reciprocal Co-IPs and IPs using 3 co-expressed proteins. We had initially shown that either ZmCRN-mCherry or CT2-YFP could IP FEA2-myc, and now we added new experiments using 3 co-expressed proteins showing that when we reciprocally IP FEA2-myc we pull down CT2-YFP and ZmCRN-mCherry (Figure 4), confirming that FEA2 indeed brings down both CT2 and ZmCRN when three proteins are co-expressed.

We also used bimolecular fluorescence complementation (BiFC) assays to independently confirm our in planta co-IPs. We used an optimized BiFC system, with monomeric Venus (mVenus) split at residue 210 to reduce background due to false positive interactions (Gookin et al., 2014). We detected YFP signal when FEA2 was fused with the N terminal part of mVenus (NmVen210) and ZmCRN was fused with the C terminal part (CmVen210) (Figure 4—figure supplement 2), confirming a direct interaction between FEA2 and ZmCRN. Similar results were reported in *Arabidopsis* using BiFC to detect CRN-CLV2 interactions (Zhu et al., 2010). However, we failed to detect a YFP signal when FEA2-NmVen210 was co-expressed with CT2-CmVen210 (Figure 4—figure supplement 2). The interaction between FEA2 and CT2 is well documented in *N. benthamiana* as well as in maize by in vivo Co-IP experiments (Figure 4, Bommert et al., 2013a). A failure to detect the same interaction using BiFC suggests that their interaction might be indirect (Figure 6). Our lab has extensively studied FEA2-CT2 interactions and repeated the in vivo co-IP many times, however in other experiments using FRET or BiFC we did not detect the interaction, suggesting the interaction is indirect, i.e. another protein or proteins bridges the interaction – we are currently working on a candidate for this bridging protein.

Lastly, as expected, no signal was detected when CT2-NmVen210 was co-expressed with ZmCRN-CmVen210 (Figure 4—figure supplement 2), confirming out co-IP results, and supporting the hypothesis that they do not interact. This is described in the new text:

“To validate these interactions, a reciprocal Co-IP experiment was carried out, in which all three proteins were co-expressed, and we again found that FEA2-Myc could IP CT2-YFP or ZmCRN-mCherry (Figure 4), further confirming that FEA2 formed complexes with both CT2 and ZmCRN. […] Lastly, as expected, no signal was detected when CT2-NmVen210 was co-expressed with ZmCRN-CmVen210 (Figure 4—figure supplement 2), confirming out co-IP results, and supporting the hypothesis that they do not interact.”

b) Role of different peptides, ZmFCP1 and ZmCLE7, on receptor complex formation. This can be done by treating/infiltrating/co-expressing each peptide to the N. benthamiana leaves along with a pair-wise combination of receptors.

As suggested, we co-infiltrated ZmFCP1 or ZmCLE7 along with the expression of FEA2-Myc, CT2-YFP and CRN-mCherry, and performed the co-IP experiment. However, adding CLE peptides didn’t affect the interaction between FEA2-myc and CT2-YFP or CRN-mCherry (Figure 4—figure supplement 3). This negative result is difficult to interpret, but we note that similar experiments have not been performed even in *Arabidopsis*, so there is no precedent to believe that the CLE ligands should lead to change in receptor complex formation. The results are described in the new text:

“Indeed, these interactions appeared to be quite stable, and they were not affected by co-infiltration of CLE peptides (Figure 4—figure supplement 3).”

Figure 4. The gel lanes appear spliced. This is not an accepted practice or data presentation.

We apologize for the oversight, and inserted lines as suggested. The western blots are now presented in the correct way.

Discussion, fourth paragraph (discussion about why clv2 phenotype is weaker than clv1). A previous report (Diévart et al., 2003) shows that missense alleles of clv1 act in a dominant-negative manner. Please include this into the Discussion.

Thanks. We added this information into the Discussion:

“This also explains why all intermediate and strong *clv1* alleles are dominant negative, as they likely interfere with the activity of other receptor kinase(s) that have functional overlap with CLV1 (Dievart et al., 2003; Nimchuk et al., 2015).”

Reviewer #3:[…] Overall this is a well written manuscript which provides some novel insight into the CLE signaling pathway involved in plant stem cell maintenance. Since, I am not a maize geneticist, I will focus my review on a) the biochemical experiments presented in this study and b) on the discussion of these novel findings:a) I understand that it is very difficult to perform any co-IP experiments in maize and that for this reason a transient system was chosen. I would find it nevertheless appropriate to at least mention in the Discussion that the outlined interactions were assayed not with native-like protein levels but rather by strongly overexpressing CRN, FEA2 and CT2.

Thanks for the suggestion, we now mentioned in the Discussion part the caveat that the interaction between ZmCRN and FEA2 was carried out in a transiently overexpression system:

“…and that FEA2 and ZmCRN interacted directly, using Co-IP and BiFC assays of proteins transiently overexpressed in N. benthamiana, suggesting that ZmCRN is a signaling component in the FEA2 pathway.”

In addition, in the revised manuscript, we provided more evidence on the interactions between ZmCRN and FEA2 using reciprocal Co-IP and BiFC in *N. benthamiana* (Figure 4 and Figure 4—figure supplement 2). These data further confirm the interaction between FEA2 and ZmCRN. For the interaction between CT2 and FEA2, we also confirmed the interaction in *N. benthamiana* by reciprocal Co-IP experiments – this interaction has been previously demonstrated using native expression in CT2-YFP transgenic maize (Bommert et al., 2013a). These new experiments are described in detail in the comments to reviewer 2, above.

b) Why are the peptide resistance assays done with such high peptide concentrations (30 μM)? Do the authors think that the binding affinities of the maize receptor complexes are much lower than in Arabidopsis? Or are there post-translational modifications missing from the synthetic peptides? Or are the peptides poorly taken up by the plant tissue?

As in our previous publications (Bommert et al., 2013a, Je et al., 2016), 5 – 30 μM peptide have been routinely used for maize root and embryo assays, respectively. Of course, we cannot know how much peptide penetrates the tissues. One reason why such a high concentration is needed might be that the synthetic peptides lack necessary post-translational modifications; in *Arabidopsis*, CLV3 is arabinosylated on a hydroxyproline modified residue (Matsubayashi, 2011), and this post-translational modification is critical for its high-affinity binding to its receptor, CLV1 (Ohyama et al., 2009). In tomato, a mutant in a hydroxyproline O-arabinosyltransferase has enlarged shoot meristems, and the mutant can be rescued with arabinosylated CLV3 (Xu et al., 2015). Thus, CLE peptides with arabinosylated hydroxyproline modifications have higher activity than the non-modified peptides.

The tri-arabinosylated CLE peptides are not commercially available, and the group that first published them were not able to provide it to us because they are very difficult to produce. However, very recently we obtained some hydroxyproline-triarabinosyl modified ZmCLE7 peptide from a collaborator (similar to ones published in Corcilius et al., 2017). We compared maize root and SAM growth following treatment with modified or non-modified peptides. As predicted, the arabinosylated ZmCLE7 peptide was more potent than the non-modified version, we estimate by ~ 10 fold. However, our collaborator could only provide a small amount of the arabinosylated CLE7 peptide and it is not enough to repeat all of the assays in the manuscript. Therefore, we propose the following modification to the manuscript:

“Although we used high peptide concentrations, the activity of CLE peptides is known to be enhanced by triarabinosylation (Ohyama et al., 2009; Matsubayashi 2011; Xu et al., 2015; Corcilius et al., 2017), and indeed we found that similarly modified ZmCLE7 peptide was about 10 fold more potent than the non-modified form (Figure 5—figure supplement 2).”

*c) Based on their findings, the authors suggest FEA2/CLV2 as a co-receptor for several CLE peptide-sensing LRR receptor kinases. In terms of the presented as well as previously established genetics on FEA2/CLV2 this would only make sense, if indeed CLV2 function can be replaced by another co-receptor* in vivo*. I would appreciate if the authors could present here a sequence alignment of the known receptor-like proteins from maize. Are there sequence homologs for CLV2 in maize (there are no obvious homologs in Arabidopsis)?*

We included a phylogenetic analysis supporting the idea that similar to CLV2 in *Arabidopsis*, FEA2 does not have any close homolog that could function redundantly in maize (Figure 6—figure supplement 1). We think the reviewer is asking why then are *fea2* phenotypes not stronger. We believe this is because of other redundant pathways, such as *fea3*, as also discussed in our response to reviewer 2.

“As a candidate receptor or co-receptor for different peptides, FEA2 does not to have any close homologs in the maize genome (Figure 6—figure supplement 1), similar to CLV2 in *Arabidopsis*, and its relatively modest phenotype may be due to compensation by partially redundant parallel signaling pathways, such as through FEA3 (Je at al., 2016) in *fea2* mutants.”